# GaN and SiC Device Characterization by a Dedicated Embedded Measurement System

**Alberto Vella** [1]**, Giuseppe Galioto** [1]**, Gianpaolo Vitale** [2],*[ID]**, Giuseppe Lullo** [1][ID]
**and Giuseppe Costantino Giaconia** [1][ID]

1   Department of Engineering, University of Palermo, Viale delle Scienze, Building 9, 90128 Palermo, Italy;
    alberto.vella@unipa.it (A.V.); giuseppe.galioto@unipa.it (G.G.); giuseppe.lullo@unipa.it (G.L.);
    costantino.giaconia@unipa.it (G.C.G.)
2   National Research Council of Italy ICAR, Institute for High Performance Computing and Networking,
    Via Ugo La Malfa 153, 90146 Palermo, Italy
*   Correspondence: gianpaolo.vitale@icar.cnr.it

**Abstract:** This work proposes a comparison among GaN and SiC device main parameters measured with a dedicated and low-cost embedded system, employing an STM32 microcontroller designed to the purpose. The system has the advantage to avoid the use of expensive laboratory measurement equipment to test the devices, allowing to obtain their behavior in operating conditions. The following KPIs (Key Performance Indicators) are measured and critically compared: threshold voltage, on-resistance and input capacitance. All the measurements are carried out in a short time interval and on a wide range of switching frequencies, ranging from 10 kHz to 1 MHz. This investigation is focused on the deviation of the figures of merit when the switching frequency changes, since it is crucial for wide-bandgap devices. The devised, low-cost, microcontroller unit allows high flexibility and system portability, while the employed equivalent-time sampling technique overcomes some issues related to the need of high sampling frequency. It allows good performances with common microcontroller embedded AD converters. To validate the proposed system, the obtained results have been compared with the time-domain waveforms acquired with a traditional laboratory oscilloscope and a study of the system's measurement errors has been carried out. Results show that GaN devices achieve a higher efficiency with respect to SiC devices in the considered range of switching frequencies. The on-resistance exhibited by GaN devices shows, as expected, an increase with frequency, which happens to switching losses, too. On the other hand, GaN devices are more sensitive to parasitic effects and the high d$V$/d$t$, due to the reduced switching times, can excite unwanted ringing phenomena.

**Keywords:** SiC; GaN; HEMT; embedded; measurement; high frequency; microcontroller; trapping effects; on-resistance; dynamic on-resistance; GaN/SiC device characterization





## 1. Introduction

The development of a switching power converter board based on wide band-gap (WBG) semiconductors brings numerous performance advantages while also opening up many issues [1]; the most important ones are related to modelling [2,3], losses in both switching [4,5] and conduction operation [6], and EMI (ElectroMagnetic Interference) effects [7]. In fact, such devices, particularly those based on GaN which allow higher switching frequency compared to traditional silicon devices, require a more detailed knowledge of the device and layout parasitics and can generate conducted and radiated emissions. With reference to switching losses, in [4,5], the lack of suitable models for estimating such losses is highlighted; in addition, measurements have to consider the probe-oscilloscope system to guarantee the fidelity in measuring the voltage–current waveforms. Measures are necessary to supply the lack of complete models; on the other hand, it is fundamental not to forget that the rise time is in the nanosecond range. As a consequence, the waveforms contain spectral component up to about 1 GHz [4], resulting particularly challenging for

the current sensor bandwidth [8] and the related converter design [9]. With reference to conduction losses, the attention is focused on the conduction resistance $R_{DS,on}$ of the devices. A frequency dependent characterization for $R_{DS,on}$ in GaN devices is proposed in [6]; it is performed by finite element simulation covering different GaN devices. Results provide evidence that there is a relevant increase in the $R_{DS,on}$ with frequencies above 1 MHz and the layout influence. The dependence of $R_{DS,on}$ on drive gate conditions is discussed in [10], where the double-pulse and multi-pulse testing methods are applied under hard switch conditions. The double-pulse method is adopted by [11] under soft switching conditions and by [12] that simplifies the current sensor using the inductor current, minimizing the power loss associated to a resistive shunt sensor. All the above cited papers highlight the importance of the device characterization in order to design a converter based on a suitable measurement system.

This paper proposes an embedded system aimed to characterize the figure of merit of GaN and SiC devices. This approach is considerably cheaper than using dedicated measurement instrumentation to suitably acquire waveforms in switching operation, since a low-cost microcontroller unit is exploited. Fast waveforms are detected by the equivalent-time sampling technique, allowing to characterize devices in a switching frequency range between 10 kHz and 1 MHz. The characterization system allows a retrieval of the $R_{DS,on}$, the threshold voltage an the input capacitance. The measurement error proved to be low, thus assessing the goodness of the method. Experimental results are proposed comparing three products: a 900 V 15 A GaN-Cascode, a 650 V 15 A GaN E-mode and a 900 V 11.5 A SiC MOSFET.

Regarding our position in relation to the literature, this paper adopts a black-box approach by using a specialized measurement system to characterize WBG devices. This system enables measuring the dynamic conduction resistance, switching losses, and threshold voltage in operating conditions, i.e., when the device under test is connected to a load and the switching frequency is varied. While manufacturers involved in the project that financed this study (the European Project GaN4AP-Gallium Nitride for Advanced Power Applications) are now investigating them, the reality of the devices' structure is not addressed here. Similar devices are evaluated for the dynamic on resistance of GaN HEMTs in [9,13,14]. In particular, [13] focuses on WBG models and applications; it discusses key concerns and describes the ways in which power devices such as SiC (diodes and MOSFETs) or GaN cascode are realized. The lower switching speed of cascode configuration, the lower $R_{on}$ of WBG devices, and significant obstacles for switching up to a few MHz are highlighted, even though no test on commercial devices is shown. In our test, we were able to verify these assertions by witnessing a decline in cascode performance as switching frequency increased. Instead, the survey [9] suggests settings for some defining factors for commercially available devices, such as the $R_{on}$. Our findings are consistent with this, and the tested devices also display a greater breakdown voltage. The issues associated with dynamic $R_{on}$ measurement are also discussed in [9], which asserts that a temporal dependence is expected and demonstrates $R_{on}$'s deterioration. We show corresponding curves versus switching frequency given at environmental temperature. Finally, [14] considers only the dynamic $R_{on}$; however, like in our approach, it is measured in operating conditions and varying the temperature as well. In addition, it provides various devices' rise and fall times, whereas we show the time-domain curve in operating conditions.

The paper is organized as follows: Section 2 provides a general outline of the considered device figures of merit; Section 3 is dedicated to a deep description of the measurement system, Section 4 offers the experimental results summarized by curves versus frequency, and in Section 5, some comparative conclusions are briefly commented on.

## 2. Wide Bandgap Power Switching Devices: Figures of Merits and Losses

Wide bandgap devices represent the most recent technology for high-frequency, high-efficiency power electronics. The higher breakdown field of a WBG semiconductor allows for devices to be employed in a wide voltage range, while the higher mobility of GaN

leads to higher switching speed and lower on-resistance. In total, the material properties of WBG semiconductors result in a device with lower switching losses than a Si device with comparable voltage and current capabilities. Therefore, silicon carbide (SiC) and gallium nitride (GaN) transistors can overcome conventional silicon power device theoretical limits.

### 2.1. GaN HEMTs Devices

Most of the GaN devices available today are lateral heterojunction field-effect transistors (HFETs), also known as high electron mobility transistors (HEMTs). These devices are typically rated at 600–650 V, although higher voltage devices will be soon available on the market due to the vertical technology structure. HFET GaN devices are intrinsically different from MOSFETs because of the inner lateral structure of the power device (Figure 1). The principal feature is the AlGaN/GaN heterojunction. At the interface between these two layers, a layer of high-mobility electrons called "two-dimensional electron gas" (2DEG) is formed as a result of the crystal polarity and of the crystal strain due to the lattice mismatch between AlGaN and GaN. The 2DEG forms a conductive channel between the source and drain of the device. Due to the presence of the 2DEG channel, the HFET is inherently a depletion-mode (normally-ON) device. This default state leads to the need for protection circuitry or soft-start circuits at the driver stage to avoid the presence of the drain-source short-circuit at system startup [11].

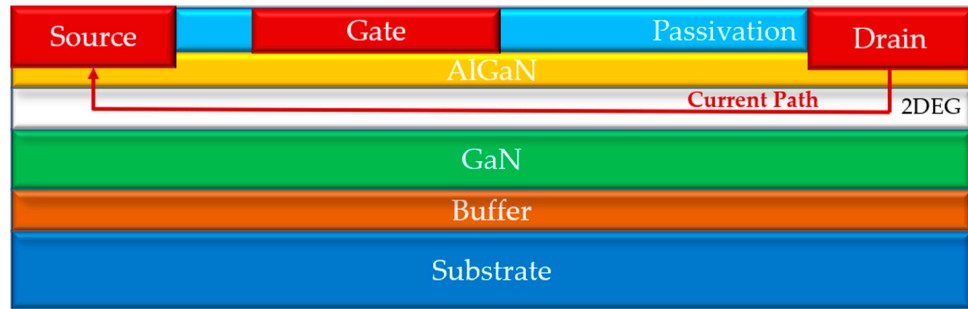

**Figure 1.** Depletion-mode lateral GaN HFET.

### 2.1.1. Cascode HEMTs

A normally OFF GaN device can be constructed using the cascode structure shown in Figure 2, which employs a depletion-mode HEMT and a low-voltage e-mode MOSFET, typically Si. The cascode structure is formed connecting the source of the Si MOSFET to the gate of the GaN HEMT and the drain of the Si MOSFET to the source of the GaN HEMT. Both devices carry the same on-state current, and the blocking voltage falls across them during the off state. The switching performances of the cascode device depend strongly on the parasitic components introduced by the connections between the two devices. This aspect can have a significant impact on power switching losses.

### 2.1.2. Enhancement-Mode Devices

Enhancement-mode devices have a structure that is similar to the structure of d-mode devices, but the main difference is that their default state is "OFF", so they are "normally-OFF" transistors. There are many ways to fabricate an e-mode GaN transistor. One way is to grow a layer of p-doped GaN onto the AlGaN barrier. The voltage generated by the positive charge of the p-doped layer is higher than the voltage generated by the piezoelectric effect of the strain. In this way, the electrons in the 2DEG are depleted and the device is off [15]. However, the threshold voltage of e-mode devices is usually low and this can bring some problems such as the ringing phenomena.

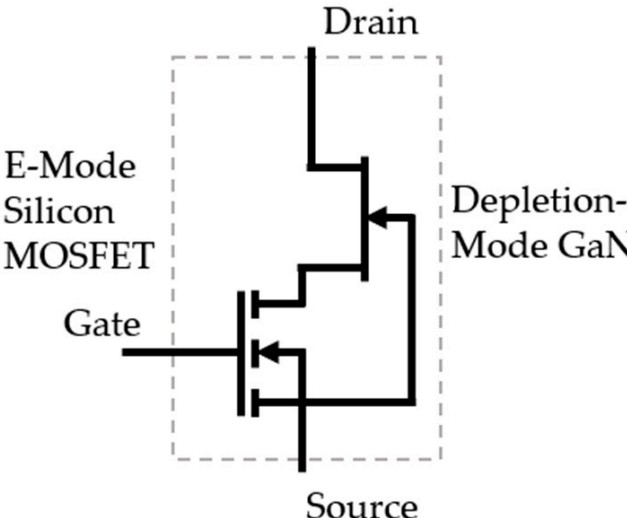

**Figure 2.** Cascode GaN HEMT made by an enhancement-mode n-channel silicon MOSFET in series with a depletion-mode GaN HEMT.

### 2.2. SiC MOSFETs

SiC devices are a competitive alternative for the next generation power electronics devices due to their medium-high voltage-current ratings. This kind of power devices has the classical structure of power MOS transistor. A significant difference can be noticed between GaN and SiC in their electron mobility, which suggests GaN to be more suitable for high-frequency applications. On the other hand, SiC has higher thermal conductivity; thus, theoretically, SiC devices could operate at higher power densities than GaN. The intrinsic high temperature capabilities of SiC power devices enable better thermal management and a reduction in the volume of heat sinks and cooling systems. When high power is a key desirable device feature, SiC semiconductors have an advantage over GaN. Moreover, SiC devices use fewer components and provide higher electric field strength, which makes it possible to achieve higher voltages with extremely low resistance and reduce the power losses [13].

### 2.3. Device Losses

Efficiency in power conversion systems is always a primary goal to achieve in order to reduce costs and enhance performance. Although realized with different materials and manufacturing processes, the most widespread power switching devices are sources of the same kind of losses, mainly consisting of switching or commutation losses, on-state or conduction losses and leakage or off-state losses [16]. This latter term can be often neglected due to the reduced drain current in off-state, in the order of some μA. Some of the main device figures of merit, which characterize their performance and must be optimized to reach the best conversion efficiency, are the on-state resistance $R_{DS,on}$, the input capacitance $C_{iss}$ and the device threshold voltage variation $\Delta V_{th}$.

Conduction losses occurring during the power device on-state can be expressed as

$$P_{on} = \frac{R_{DS,on} I_d^2 t_{on}}{T_s}, \tag{1}$$

where $R_{DS,on}$ represents the on-resistance of the switching device, $I_d$ is the on-state current, $t_{on}$ is the on-state conduction time and $T_s$ the switching period. Therefore, in order to minimize conduction losses, a power switching device with a low $R_{DS,on}$ must be chosen. Due to the most recent available power device technologies, such as Silicon Carbide and Gallium Nitride, it is possible to lower the on-resistance and achieve higher performance with respect to conventional Silicon device theoretical limit.

It should be remarked that, unlike in traditional Si-based devices, the parameter $R_{DS,on}$ shows a variation with the frequency and its dynamic value is different from the one measured in static condition. In addition, it can be influenced by the layout, requiring a measurement "in situ" to correctly identify the device model.

On the other hand, switching power losses occur during the transitions between on and off states, or, equivalently, when the drain current $I_d$ and the drain-source voltage $V_{ds}$ are different from zero at the same time. Therefore, the mathematical expression for these losses is

$$P_s = \frac{1}{2} V_{ds} I_d f_s \left( t_{c(on)} + t_{c(off)} \right). \tag{2}$$

In this expression, $f_s$ is the switching frequency, while $t_{c(on)}$ and $t_{c(off)}$ are the on and off switching times. The main figure of merit that affects power switching losses is the device input capacitance, which must be minimized in order to reduce the switching times and so the switching power losses.

The Equation (2) defines the worst case for switching losses; it is derived under the hypothesis that, as an example, during the turn-off, the conduction current remains constant until the voltage $V_{ds}$ reaches its maximum value, and then $I_d$ start falling. Aiming to optimize the converter design including the cooling system, the value of $P_s$ needs a precise calculation that can be retrieved only by an accurate model simulation or directly by measurements.

Finally, power device reliability can be strongly affected by threshold voltage instability. A negative threshold voltage shift can bring to unwanted turn-on of the device, while a positive shift can make switching times longer and affect the on-resistance, so this phenomenon can increase the device's power losses, leading to larger conduction losses.

It is worth to characterize these key device figures of merit as functions of switching frequency in order to describe the behavior of these devices in a wide range of operating frequencies. The embedded measurement system presented in this work provides a low-cost portable solution with respect to available commercial products [17,18]. Beside this, it must be pointed out that the maximum frequency for the capacitance measurement normally reaches 10 MHz, while in the present work, a maximum of 1 MHz can be currently reached, even if this limit may be increased in future developments. Finally, due to the equivalent time sampling technique, the developed embedded platform can achieve a high equivalent sampling frequency with a simpler and cheaper design while keeping a high grade of flexibility [19].

## 3. Embedded Measurement System

The measurement system introduced in this work is suitably designed to characterize the previously described figures of merit. The microprocessor unit is encompassed in a board able to contain the device under test and related components. The developed system is able to measure the gate voltage, the drain voltage and the drain current during switching operations and in a wide range of switching frequencies. In particular, the selected frequency range is between 10 kHz and 1 MHz, allowing to investigate the behavior of high frequency devices such as the Gallium Nitride HEMTs.

In order to meet the sampling and measurement requirements, an STM32H743ZI microcontroller based on the high-performance Arm® Cortex®-M7 32-bit RISC core has been selected due to its embedded high resolution timer (HRTIM) capable of reaching 480 MHz of internal clock frequency. The Cortex®-M7 core features a floating point unit (FPU) which supports Arm® double-precision (IEEE 754 compliant) and single-precision data-processing instructions and data types. STM32H743 devices incorporate high-speed embedded memories with a dual-bank Flash memory of up to 2 Mbytes, up to 1 Mbyte of RAM (including 192 Kbytes of TCM RAM, up to 864 Kbytes of user SRAM and 4 Kbytes of backup SRAM), as well as an extensive range of enhanced I/Os and peripherals connected to APB buses, AHB buses, $2 \times 32$-bit multi-AHB bus matrix and a multi-layer AXI interconnect supporting internal and external memory access.

All the devices offer three ADCs, two DACs, two ultra-low power comparators, a low-power RTC, a high-resolution timer, 12 general-purpose 16-bit timers, two PWM timers for motor control, five low-power timers, and a true random number generator (RNG). The devices support four digital filters for external sigma-delta modulators (DFSDM). They also feature standard and advanced communication interfaces. As already mentioned, this embedded platform includes three independent ADCs. The ADCs are successive approximation (SAR) converters. One relevant feature is the possibility to use a self-calibration function in order to compensate offset and non-linearity errors. This operation significantly reduces the measurement error and allows, above all, the measurement of very low voltages such as the drain-source voltage during conduction; that is one of the targets of the designed system.

It should be remarked that the same microprocessor unit could allow the implementation of additional software dedicated, as an example, to the converter control or to real-time diagnostic. From the knowledge of the device parameters, an online optimization can be performed during operation.

In Figure 3, the structure of the measurement system is shown. A picture of the realized embedded board is given in Figure 4. The load power resistor, whose value depends on the desired output current (150 Ω for GaN devices, 25 Ω for SiC device), is employed both to limit the drain current and to sense the drain voltage. The gate resistor selection is crucial to set the gate equivalent time constant and also the switching speed of the power device. The gate voltage sensing circuit is essentially a level shifter used to adapt the voltage levels to the ADC voltage reading Input range.

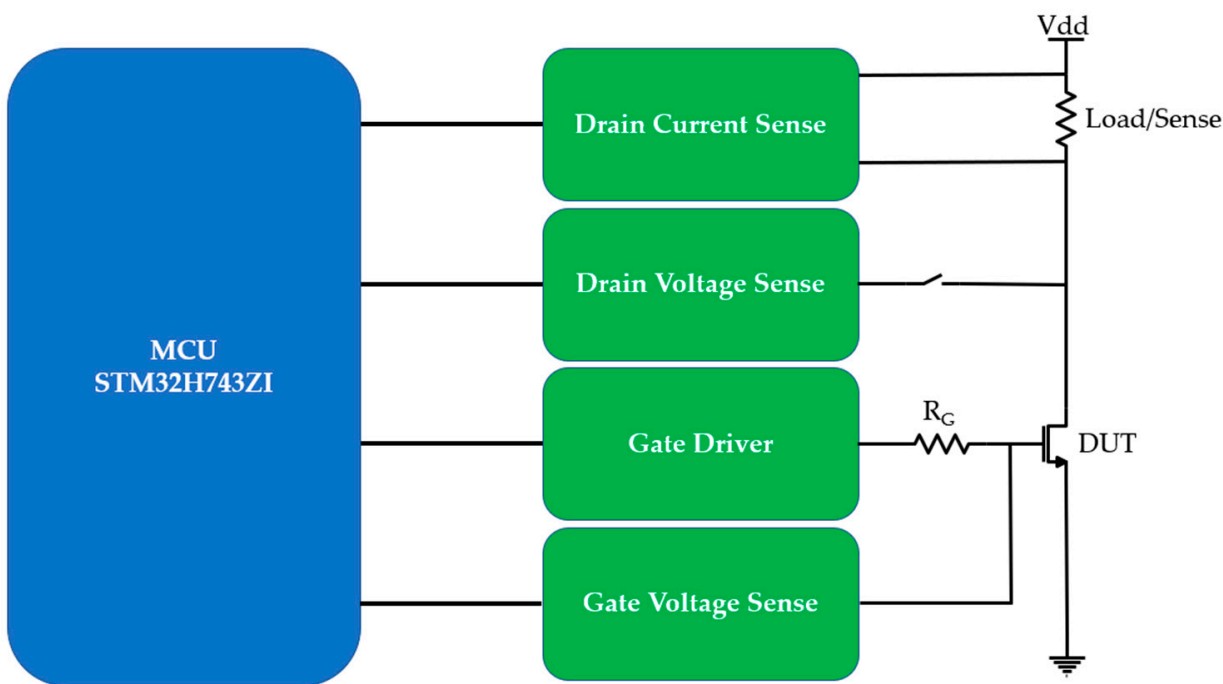

**Figure 3.** Measurement system block diagram.

The measurement system employs the *constant current* method to determine the threshold voltage, as described in [18]. With this method, the threshold voltage is calculated as the gate voltage corresponding to a predefined drain current. This constant drain current, $I_{d,c}$, is selected considering the current measurement uncertainty, $u_i$, and its value was chosen equal to 100 times $u_i$ to avoid noise corruption. In order to calculate the threshold voltage variation in one period ($\Delta V_{th}$), the device is switched with different duty cycles and the threshold voltage is calculated for every duty-cycle value, obtaining the variation in one period, as explained in [19].

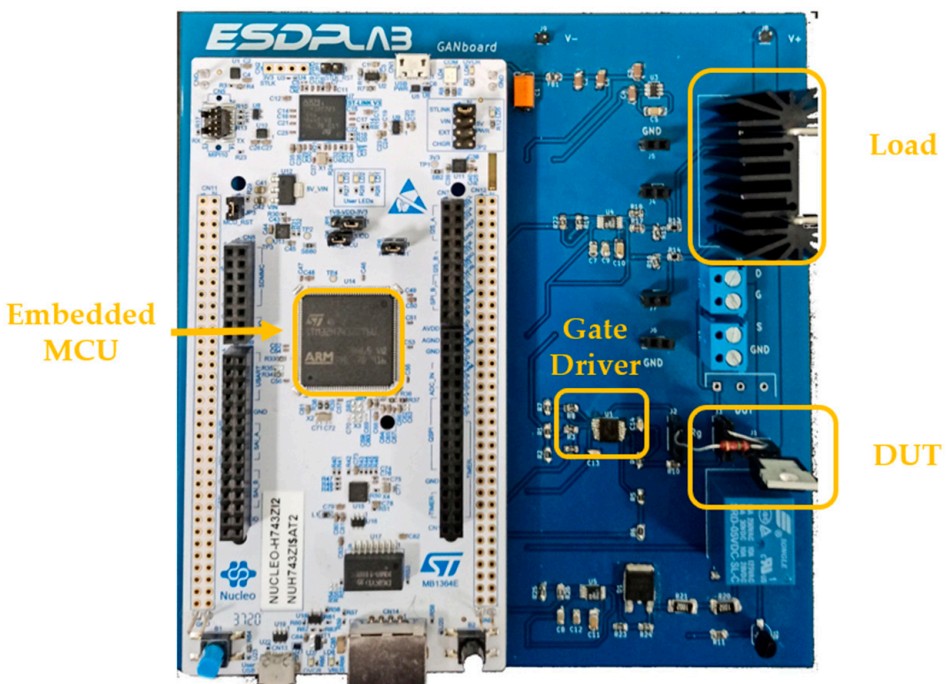

**Figure 4.** Embedded measurement system board: Embedded MCU, Gate Driver, DUT with gate resistance, Load.

The drain-source on-resistance, $R_{DS,on}$, is measured as the ratio between the drain voltage $V_D$ and the drain current $I_D$. A gate signal with 90% duty cycle switches the device and, during conduction state, an average value of $V_D$ and $I_D$ is calculated to carry out the on-resistance calculation.

Next, the input capacitance $C_{iss}$ of the considered switching power device is derived from the selected gate resistor and the equivalent gate time constant related to the gate voltage rising time. This relation can be expressed as

$$C_{iss} = \frac{\tau}{R_G}, \tag{3}$$

where $\tau$ is the time constant related to the gate voltage transition and $R_G$ the external gate resistance. The time constant is calculated as the time interval between the start time of the rising transient of the gate voltage and the time instant when it reaches the 63% of the total voltage span.

Traditional sampling techniques with high-speed and high-frequency switching devices would require an ADC with a high sampling frequency; albeit technically feasible, it would result in a significant rise in system costs. Therefore, in the presented system, the equivalent time sampling technique has been implemented. With this technique, a periodic signal can be sampled with a reduced sampling frequency without losing information. This technique consists in sampling the considered signal with a sampling period equal to the signal period plus an additional time equal to

$$T_{ET} = \frac{T}{n}, \tag{4}$$

where $T$ is the period of the input signal and $n$ is the desired number of samples. Therefore, the sampling period can be expressed as

$$T_S = T + T_{ET}. \tag{5}$$

With this technique, a sampling frequency slightly lower than the signal frequency can be used to make the measurement system simpler and cheaper. In order to further enhance the conversion performance, the ADC is triggered by the high-frequency clock HRTIM and the data are transferred employing the embedded Direct Memory Access (DMA) module, reducing the computation time and avoiding interrupt routines. The accuracy of the equivalent time sampling technique depends on the precision of the sampling time, where the quantity $T_{ET}$ must be maintained constant and with a negligible error. In our case, the error on $T_{ET}$ is estimated in about 2.5 ns. The accuracy has also been experimentally verified.

### 3.1. Hardware Implementation

The driving circuits of the measurement setup are briefly described here. For the drain current sensing circuit, as shown in Figure 5a, two voltage dividers are applied at both terminals of the sense resistor to protect the operational amplifier, working in a differential configuration, from the high voltage of the DUT's power supply. A secondary effect of the voltage divider is to increase the input resistance of the amplifier: the chosen set of resistors results in a differential input resistance well above 1.2 MΩ, much larger than the sense resistor, thus minimizing the measurement errors. The total reduction factor between the voltage across the sense resistor and the "TO MCU" output is equal to 0.045.

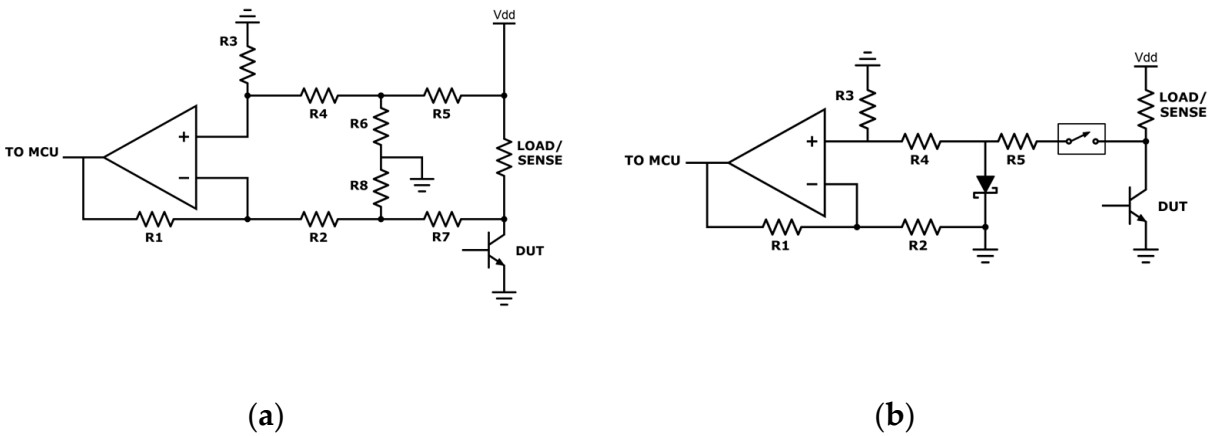

**(a)** **(b)**

**Figure 5.** Drain sensing circuits: (**a**) drain current sensing circuit; (**b**) drain voltage sensing circuit.

In Figure 5b, the drain voltage sensing circuit is shown. Here, the main issue when measuring the DUT's drain voltage is its large difference between the off-state and on-state. As we are interested only in the drain voltage value during the on-state ($R_{DS,on}$ can be calculated from this value), instead of adopting a large voltage attenuation that would make the drain voltage in on-state indistinguishable from noise, a voltage clamp circuit was used. It is constituted by resistor *R5* and a fast SiC Schottky diode. During off-state, the Schottky diode clamps the amplifier input voltage to its knee voltage, a few hundred mV. During the on-state, the drain voltage maintains below the Schottky threshold voltage and can be correctly measured. Consequently, in the off-state, the amplifier input voltage is the diode threshold and it is not considered, while in conduction state, the true drain voltage can be sampled. As can be seen from the schematics, a relay is connected between this stage and the DUT's drain in order to disconnect the drain when drain voltage measurements used for the extraction of the on-resistance are not performed. This choice is due to the fact that the diode stray capacitance can affect the drain voltage rising and falling time used for the extraction of the input capacitance. In fact, the adopted circuit introduces a minor attenuation of 0.976 that can be considered during data conditioning.

In Figure 6a, the gate driver is shown. It is a non-inverting summing amplifier with the aim of shifting the input level from the MCU to the driving voltage range of the DUT.

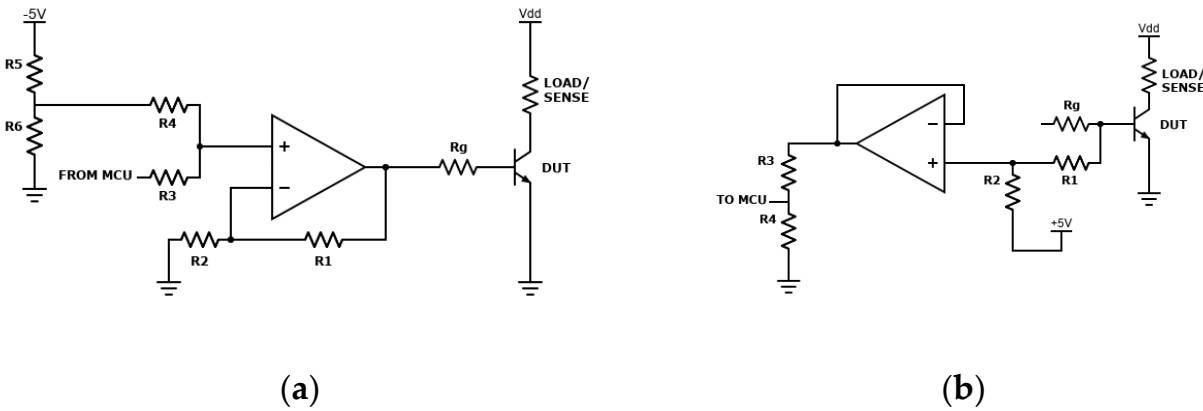

**(a)**  **(b)**

**Figure 6.** Gate circuits: (**a**) gate driving circuit; (**b**) gate voltage sensing circuit.

Finally, in Figure 6b, the gate voltage sensing circuitry is presented. Here, an operational amplifier is employed in non-inverting unity-gain summing configuration. The output voltage is fed to the MCU with a simple voltage divider [19].

*3.2. Firmware Implementation*

The measurement system firmware consists of a finite states machine formed by five different states. The system first enters once only into the calibration state, then it can run the other four states in looping sequence.

As it can be seen from the flow chart represented in Figure 7, after the initialization state, the calibration state starts. This state exploits the ADCs feature of self-calibration present in the HAL libraries in order to increase the measurement accuracy.

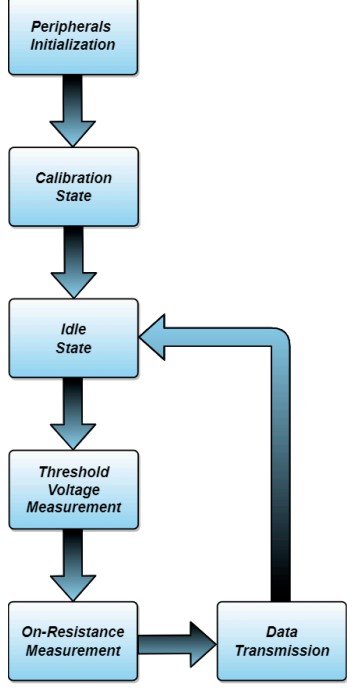

**Figure 7.** Firmware Finite State Machine (FSM).

The following state is the idle state, allowing the microcontroller in a rest state until the user push-button starts the interrupt routine. During the idle state, a user LED flashes to indicate the current state. Moreover, a counter variable, used to enumerate the sampling instants, is set to zero. Once the user button is pushed, the LED is turned off and the state is

switched to the next state. In the two following states, the signals used for threshold voltage, input capacitance and on-resistance measurements are sampled. When the sampling process expires and both the threshold voltage measurement state [20,21] and the on-resistance measurement state [22] are executed, the system steps to the next, the data transmission state. Here, the sampled signals are sent through the USART to the PC, where they are decoded and processed using MATLAB.

## 4. Results

The experimental tests were performed on the following devices: a 900 V 15 A-rated GaN-Cascode, a 650 V 15 A-rated GaN E-mode and a 900 V 11.5 A-rated SiC MOSFET (products at the end of the production cycle).

The measurement frequency for GaN devices extends to 1 MHz, while for SiC MOSFET it is equal to 200 kHz due to SiC devices' stress frequency limit. Moreover, the bias conditions of the tested devices are different: for GaN Cascode and E-mode, the supply voltage $V_{dd}$ is 60 V, and the operating current $I_d$ is 0.4 A, while for SiC MOSFET, they are 50 V and 2 A, respectively. This is due to the higher bias current level needed by SiC devices to work in saturation region. For each figure of merit, uncertainty is provided.

In Table 1, the absolute measurement uncertainties of the considered quantities are shown. The absolute uncertainties are calculated as

$$u = \sqrt{\sum_i \left(\frac{df}{dx_i}\right)^2 u_{x_i}^2} \, , \tag{6}$$

where $f$ is the function that defines the measurement, $x_i$ are the measured quantities, $u_{x_i}$ is the absolute uncertainty related to each single quantity, while $u$ is the calculated absolute uncertainty. As an example, the absolute uncertainty of the on-resistance measurement is described below. Since the expression of the resistance is $R = V/I$, the uncertainty is calculated as follows:

$$u_R = \sqrt{\left(\frac{1}{I}\right)^2 u_{V_D}^2 + \left(-\frac{V}{I^2}\right)^2 u_{I_D}^2} = 0.022 \text{ m}\Omega \, , \tag{7}$$

where $u_{V_D} = \frac{q}{2\sqrt{3}}$ is the drain voltage measurement uncertainty, $u_{I_D} = \frac{q}{2\sqrt{3}} \cdot k$ is the drain current measurement uncertainty, $k$ is the coefficient converting the voltage across the sensing resistor into the drain current, and $q = 3.3/2^{16}$ is the voltage quantization error, since a 16-bit ADC with a 3.3 V reference voltage has been used.

**Table 1.** Measurement absolute uncertainty of the device figures of merit.

| Figure of Merit | Uncertainty | Type of Uncertainty |
|:---:|:---:|:---:|
| $C_{iss}$ | 35 pF (worst case) | Absolute |
| $R_{DS,on}$ | 0.022 mΩ | Absolute |
| $\Delta V_{th}$ | 0.007% | Relative |

For the threshold voltage variation characterization, the relative uncertainty has been calculated as

$$u = \sqrt{\sum_i u_{x_i}^2}. \tag{8}$$

In the following figures, the experimental results of the measurement system are presented.

In Figure 8, the switching gate and gate driver voltage measured at 100 kHz with a fast Tektronix MSO56 oscilloscope are shown.

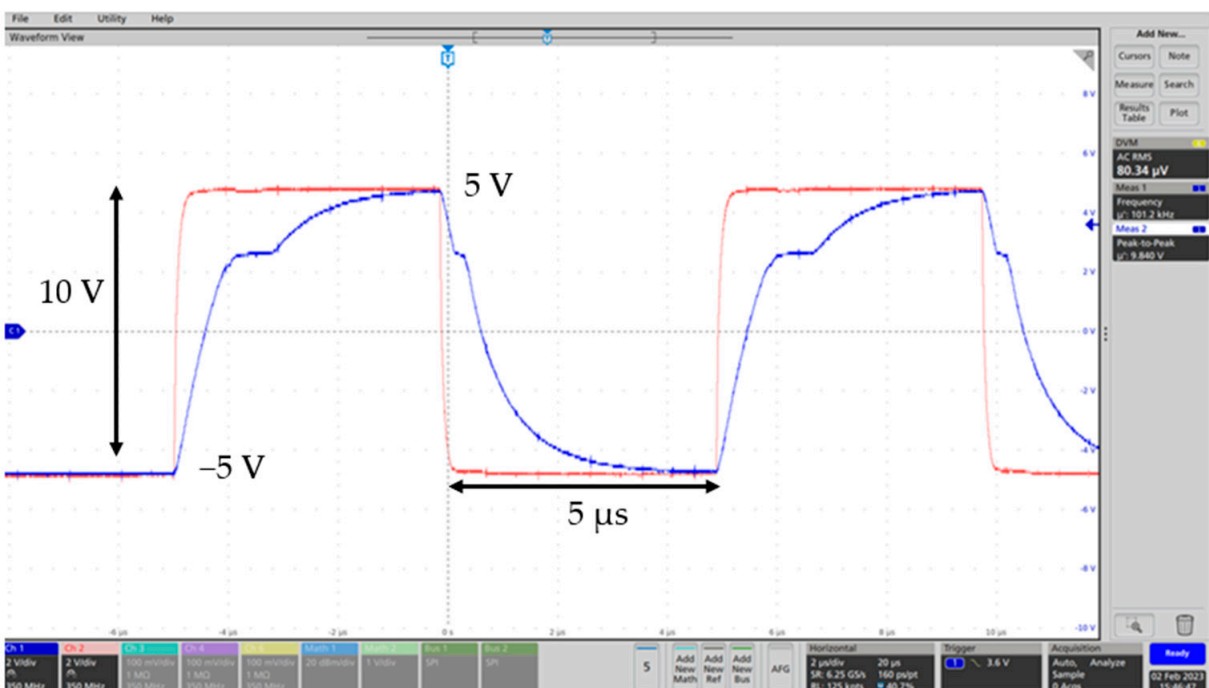

**Figure 8.** Measured gate (blue) and gate driver (red) voltage transitions at 100 kHz switching frequency with a 50% duty cycle. Peak-to-peak voltage for gate driver voltage is 10 V.

In Figure 9, a zoom for switching gate (red) and drain (blue) transition at 100 kHz is shown. It is possible to distinguish two time intervals in the drain falling transition, the first lasting almost 125 ns and the next lasting almost 600 ns.

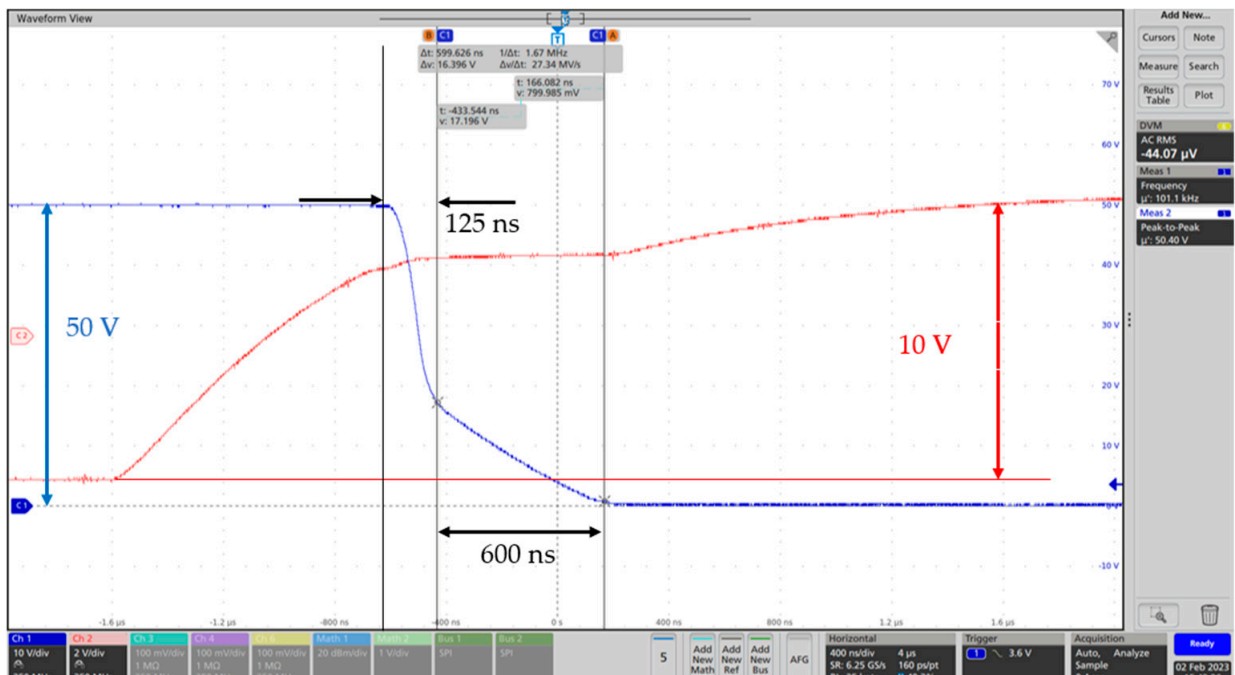

**Figure 9.** Zooming of switching gate (red) and drain (blue) voltage transitions at 100 kHz.

In order to verify the goodness of the time equivalent sampling system, a comparison between the gate voltage measured with the oscilloscope and the one sampled by the embedded measurement system was performed. The obtained waveforms are shown in

Figure 10. As can be seen from the two waveforms, the measurement system is able to accurately sample the signals with the equivalent time sampling technique. Moreover, in Figure 11, the Fast Fourier Transform (FFT) of the switching drain voltage is shown. Since an external gate resistor is employed, it could affect the behavior of the DUT's working operation. The presence of high even harmonics can be noted, as in Table 2, due to the influence of the gate resistor that transforms a pure square wave into an exponential decay, which must be considered in the design of a standard-compliant EMI filter.

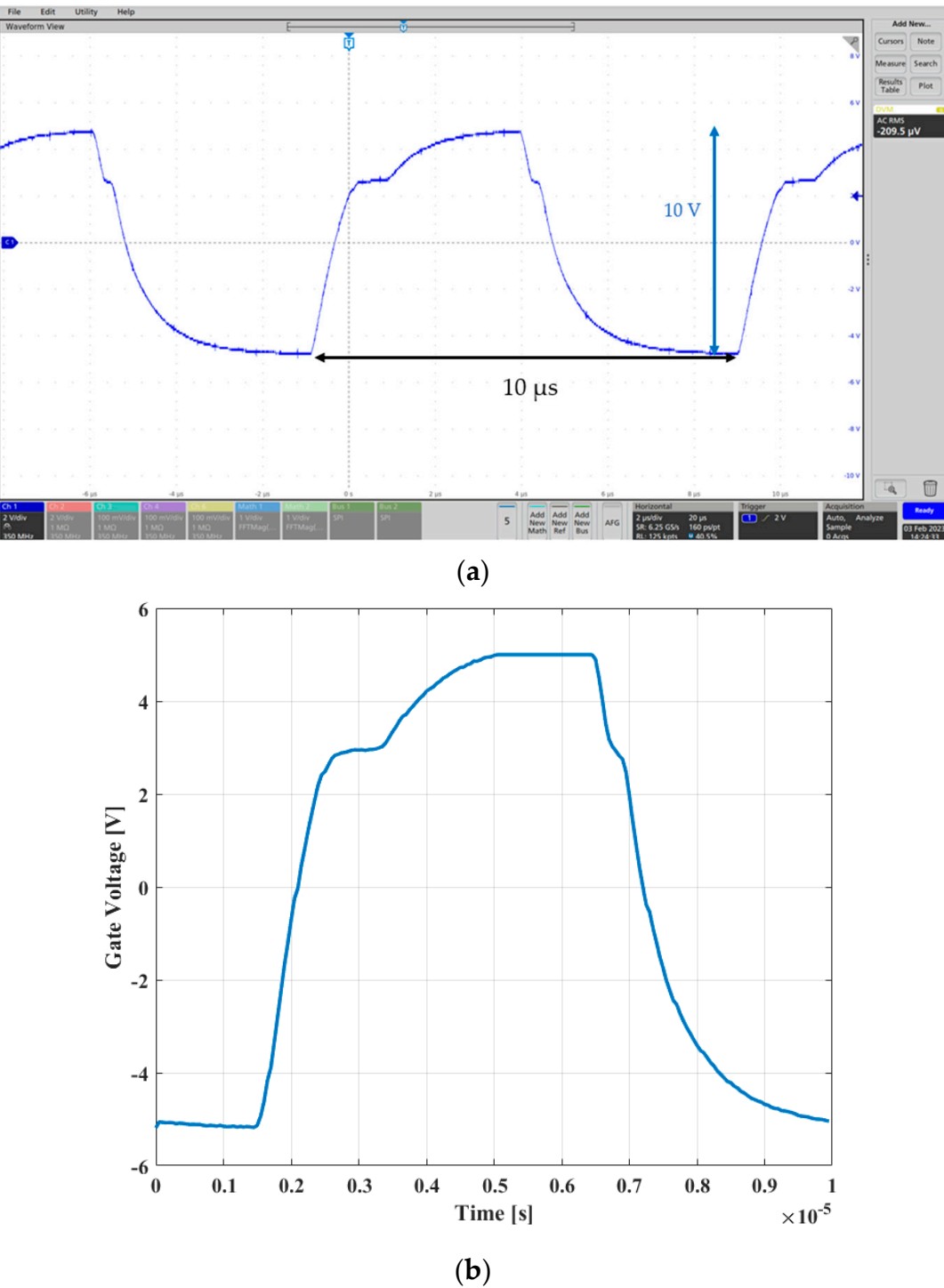

(**a**)

(**b**)

**Figure 10.** Switching gate voltage at 100 kHz. (**a**) Oscilloscope measurement. (**b**) Waveform obtained by the embedded measurement system.

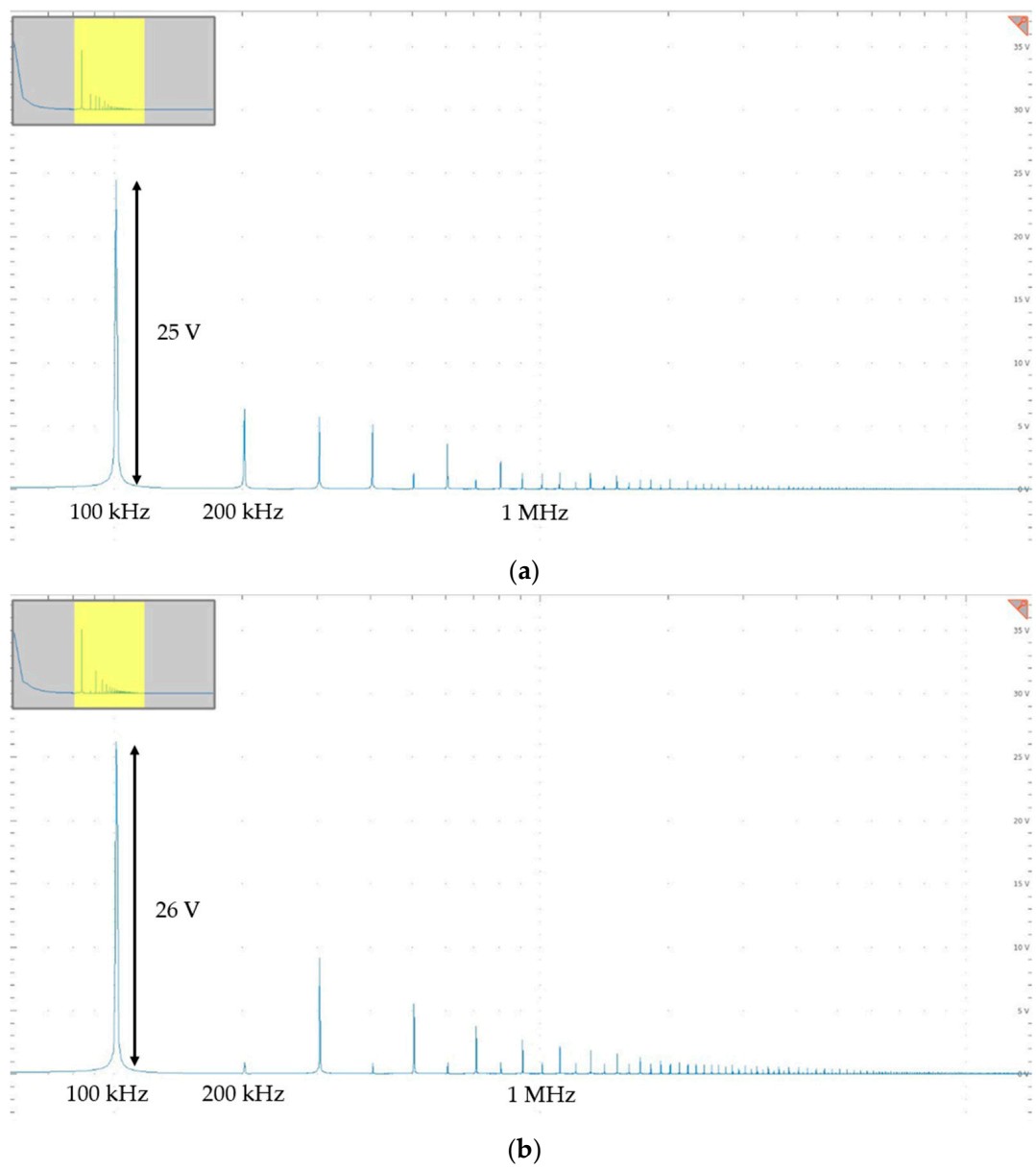

**Figure 11.** Drain voltage FFT at 100 kHz switching frequency: (**a**) with gate resistance; (**b**) w/o gate resistance.

**Table 2.** Drain voltage FFT data at 100 kHz switching frequency and 60 V supply voltage.

| Peak Frequency | Peak Amplitude w/ Gate Resistor | Peak Amplitude w/o Gate Resistor |
|:---:|:---:|:---:|
| 100 kHz | 25 V | 26 V |
| 200 kHz | 8 V | 1 V |
| 300 kHz | 7 V | 10 V |
| 400 kHz | 6 V | 1 V |
| 600 kHz | 4 V | 1 V |
| 800 kHz | 2 V | 1 V |

In Figure 12, the approximation of gate voltage charging process is represented. The time constant is calculated as explained in Section 3 and employed to calculate the exponential charge of the gate capacitance as

$$V_c = V_p \left( 1 - e^{-\frac{t}{\tau}} \right), \tag{9}$$

where $V_c$ is the capacitor voltage and $V_p$ is the steady-state voltage. It can be noted from Figure 12 that the approximation with the previous calculation follows well the real average behavior of the gate voltage during the charging transient.

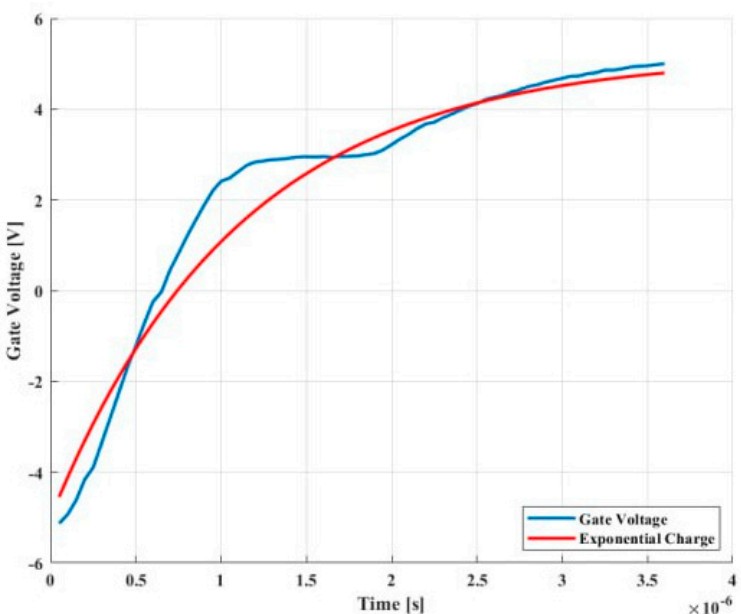

**Figure 12.** Exponential charge approximation of the rising gate voltage.

As shown in Figure 13, the input capacitance $C_{iss}$ (measured with Equation (3)) of the three considered devices almost doubles the low frequency value in the considered range of frequencies. This is due to electron trapping phenomenon occurring within the device switching period.

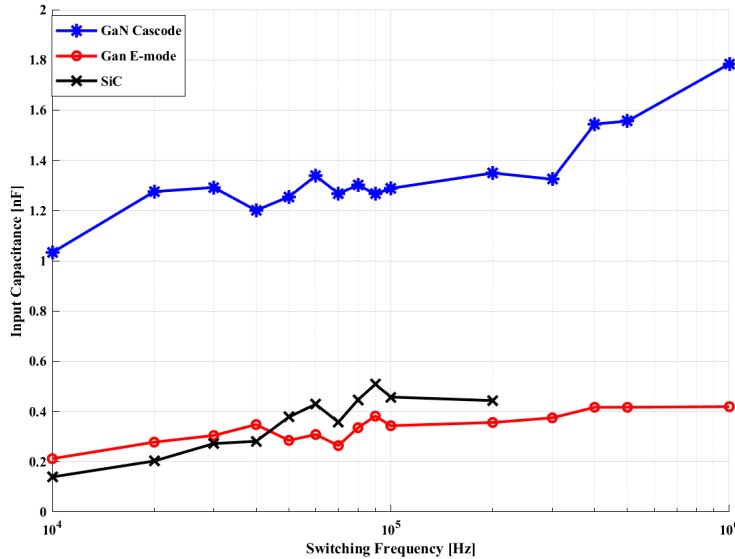

**Figure 13.** Input capacitance vs. switching frequency with Equation (3).

In order to further characterize the reproducibility of the measurements, a variance analysis over 100 measurements was carried out for the three KPIs ($C_{iss}$, $R_{DS,on}$, $\Delta V_{th}$). The experimentally obtained results show a measurement variance of $2 \times 10^{-15}$ for the gate equivalent time constant, $1.18 \times 10^{-6}$ for the $R_{DS,on}$, and 0.0031 for the $\Delta V_{th}$. By comparing these variances with their correspondent KPIs' mean values, an overall measurement error below 2% has been observed in the worst case. To experimentally evaluate whether the equivalent time sampling technique was a good measurement method, an initial test was performed aiming to answer the following question: how much error will be accumulated if a magnitude related to KPIs is sampled at a fixed frequency (thus removing the $T_{ET}$ term from the expression (5))?

Then, the DUT's gate voltage was sampled many times with a 1 MHz fixed frequency, while the exact same frequency was used as the DUT's switching frequency. Under these circumstances, the system was capable to accumulate gate voltage samples exactly one period apart, theoretically equal to each other. In fact, these samples were affected by errors and their variance was equal to $1.05 \times 10^{-4}$, while the mean value of the gate voltage samples was 3.53 V, hence experiencing an overall error of the proposed measurement method well below 1‰.

In Figure 14, the $R_{DS,on}$ behavior as a function of the switching frequency is shown. The measured values are referred to the on-resistance at a switching frequency of 10 kHz. This choice is motivated by the difference in the $R_{DS,on}$ values among the devices under test, allowing for a better visualization of the frequency behavior. In SiC devices, the variation is minimal, while GaN devices show a more consistent increase in the high-frequency region, approximately 25–30% at 1 MHz. $R_{DS,on}$ reference values at 10 kHz are the following: 408.4 mΩ for SiC, 165.8 mΩ for GaN E-mode and 163 mΩ for GaN Cascode.

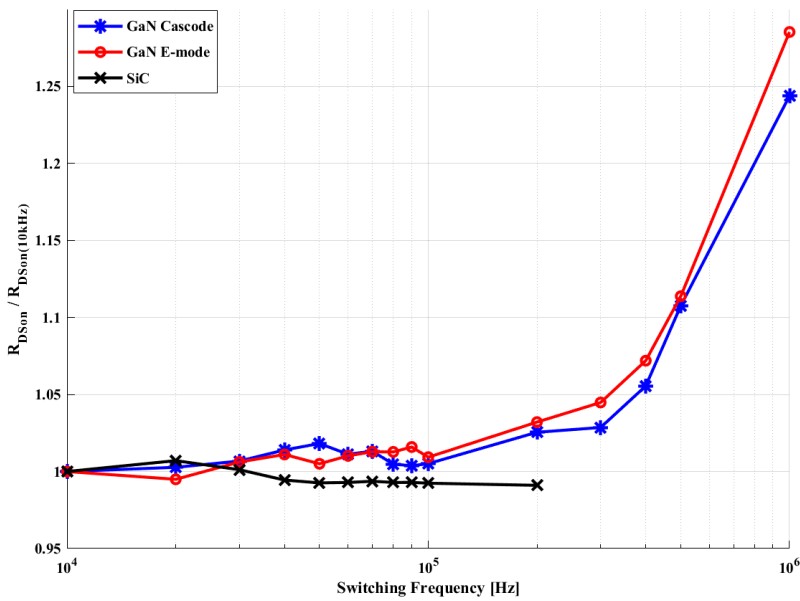

**Figure 14.** $R_{DS,on}/R_{DS,on\ (10\ kHz)}$ vs. switching frequency.

The last considered figure of merit is the threshold voltage variation during a switching period as a function of the switching frequency as in Figure 15. The post-processed values of $\Delta V_{th}$ are calculated as the difference between the positive threshold voltage variations during the on-state and the negative ones during the off-state. GaN cascode shows an almost null variation of its threshold voltage as a function of switching frequency. GaN E-mode's threshold voltage variation increases at high switching frequencies of hundreds of mV. Finally, the highest variation occurs for the SiC device, whose threshold voltage increases over 4 V at increasing switching frequencies [19].

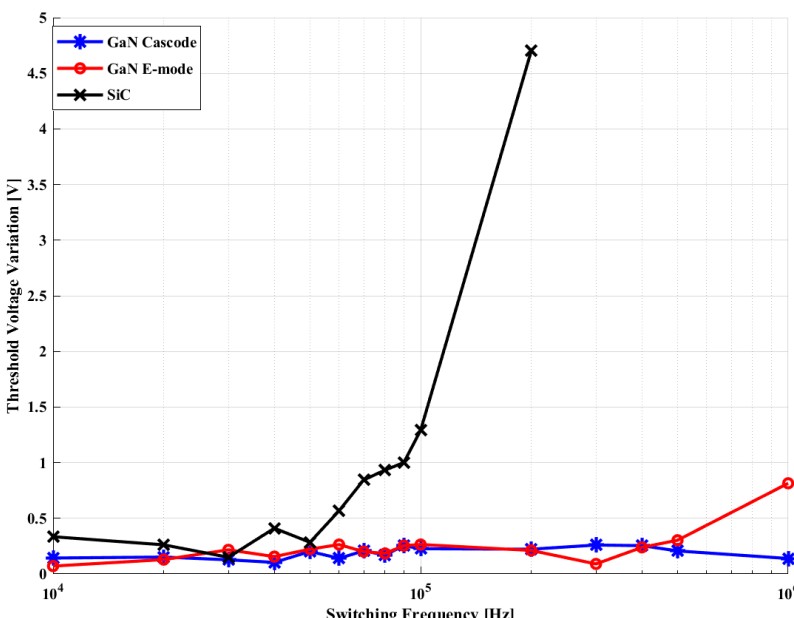

**Figure 15.** $\Delta V_{th}$ vs. switching frequency.

As mentioned before, power switching losses are crucial to determine the overall power conversion system efficiency, since they are the main contribution to the total losses. Here, these losses are quantified as in (2) from the instantaneous values of drain voltage and current. In Figure 16, the switching losses as a function of frequency are presented. These were calculated during the turn-off transition since during the turn-on the clamping diode losses affect the calculation. It can be noted that SiC devices show slightly higher losses in their operating range of frequencies. On the other hand, GaN HEMTs losses show a similar trend, remaining linear with the switching frequency up to about 500 kHz, and they have a strong increase at higher frequencies.

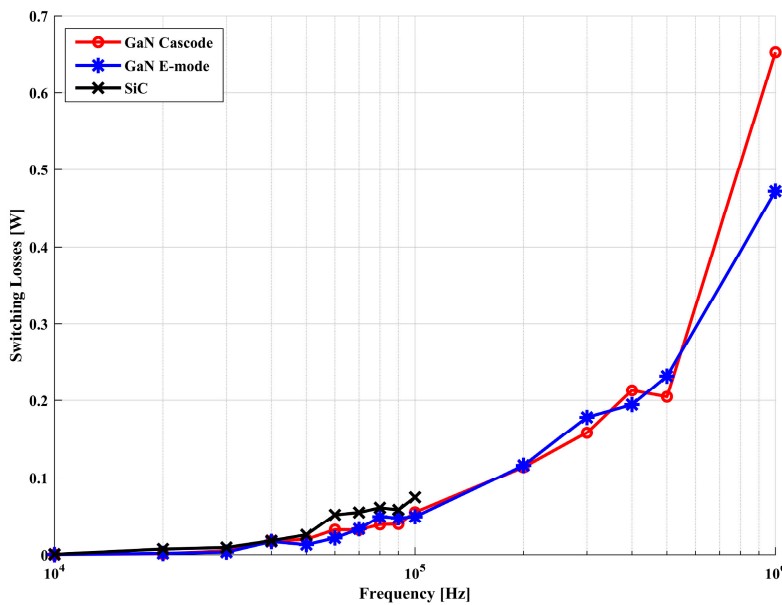

**Figure 16.** Power switching losses vs. frequency.

As a last remark, a measurement test with an output capacitance between drain and source was carried out in order to characterize the $R_{ds,on}$ behavior. The test with 50 pF capacitance shows almost negligible influence on the obtained values of $R_{ds,on}$.

## 5. Conclusions

A measurement system based on a low-cost microprocessor board was presented in this paper. It allows to retrieve the main parameters of wide-bandgap devices without employing expensive measurement systems. The board architecture was described in detail. Its features were verified on three devices (a SiC, a GaN and a cascode-GaN) mounted on the board and tested in real operating conditions. Since the frequency behavior of the extracted figures of merit physically resides in the electron trapping and de-trapping times, the tests were performed varying the switching frequency, up to 1 MHz for GaN switches. The proposed measurement system was validated and the measurement error assessed. Results confirm the results obtained in the literature with traditional measurement systems. In fact, for GaN devices, the electron trapping time is orders of magnitude lower than the de-trapping time as in [22]. With increasing frequency, both the off-state and on-state times are reduced, but since the trapping time is much smaller, this does not affect the electron trapping but only the electron de-trapping. This leads to a bigger amount of trapped electrons at higher frequencies and subsequently to an increase in the on-resistance and threshold voltage variation as in [23].

For SiC devices, during off-state electrons tunnel out of the oxide causing a negative shift of $V_{th}$. During on-state, electrons can tunnel back into the oxide, causing a positive shift of $V_{th}$ confirming the results of [20]. Since the device is driven with 0–15 V gate voltage, the positive variation is much higher than the negative one, leading to a higher positive $\Delta V_{th}$.

The present work shows the behavior of the most relevant figures of merit of GaN and SiC power devices. The study is still under progress to deepen the involved phenomena and their causes, still under investigation also in the literature [19,24,25]; however, it provides a reliable basis for characterizing and modelling new wide-bandgap devices.

**Author Contributions:** Conceptualization, A.V., G.G., G.L., G.C.G. and G.V.; methodology, A.V., G.G., G.L., G.C.G. and G.V.; software, A.V., G.G. and G.C.G.; validation, A.V., G.G., G.L., G.C.G. and G.V.; formal analysis, A.V., G.G., G.L., G.C.G. and G.V.; investigation, A.V., G.G., G.L., G.C.G. and G.V.; resources, A.V., G.G., G.L., G.C.G. and G.V.; data curation, A.V., G.G. and G.C.G.; writing—original draft preparation, A.V., G.G., G.L., G.C.G. and G.V.; writing—review and editing, A.V., G.G., G.L., G.C.G. and G.V.; visualization, A.V., G.G., G.L., G.C.G. and G.V.; supervision, G.L., G.C.G. and G.V.; project administration, G.C.G.; funding acquisition, G.C.G. All authors have read and agreed to the published version of the manuscript.

**Funding:** This work has been carried out in the framework of the European Project GaN4AP (Gallium Nitride for Advanced Power Applications). The project has received funding from the Electronic Component Systems for European Leadership Joint Undertaking (ECSEL JU), under grant agreement No. 101007310. This Joint Undertaking receives support from the European Union's Horizon 2020 research and innovation programme, and Italy, Germany, France, Poland, Czech Republic, Netherlands.

**Data Availability Statement:** The data used in this study has been extracted from the list of references and the most relevant are reported in the figures and tables of the paper.

**Conflicts of Interest:** The authors declare no conflict of interest.

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
