# Peer review of "GaN and SiC Device Characterization by a Dedicated Embedded Measurement System"

_electronics, doi:10.3390/electronics12071555_

Round 1
Reviewer 1 Report
This manuscript presents a comparison of the main parameters of GaN and SiC devices, which were measured using a dedicated and low-cost embedded system designed for this purpose and employing a STM32 microcontroller. There are a few typos, such as line 95: “cascode structure shown in Figure 7.” Once corrected, the manuscript should be published as it is.
Reviewer 2 Report
In the presented article the authors report a development of a microcontroller-based measurement system for power switching devices characterization. Measurements of threshold voltage, on-resistance and input capacitance have been conducted for GaN enhancement-mode HEMT, GaN Cascode HEMT, and SiC MOSFET devices. Below are the points that should be addressed to improve the quality of the article:
1) It will be useful to add more information about the tested devices. The device characteristics depend on geometric dimensions and fabrication technology. It will be also useful to compare the measured characteristics with existing literature data.
2) Description of the error analysis on the page 10 is unclear. Does the equation 6 express the uncertainty of a function of several variables? What is the meaning of the index i in the equation 7? Is a square root sign missing in the equation 7? What is the difference between the measured mean values of the considered quantities for the different measuring systems?
3) Copyright permission may be need for the figures 1 and 2, which are taken from the refs 11 and 12.
4) Page 1, line 34. The abbreviation "EMI" appears without definition.
5) Page 3, line 95. There is a misprint in the figure number.
6) In the figures 8,9 10a and 11 the font size for axis labels is very small.
7) Are there different voltage scales for switching gate and drain voltages in the figure 9?
8) In the figure 10b there is a misprint in the label "Gate Voltage".
